# Feasibility of Using Patient Portal to Enhance Patient Engagement and Patient-Generated Data in Tertiary Hospital

**DOI:** 10.3390/healthcare13050518

**Published:** 2025-02-27

**Authors:** Ghaliah H. Alfurayh, Abdullah T. Alanazi, Hanin S. Aldalham

**Affiliations:** King Faisal Specialist Hospital & Research Centre, Riyadh 12713, Saudi Arabia

**Keywords:** patient portal, patient engagement, patient-generated data, personal health records

## Abstract

Objectives: This study aimed to evaluate the experiences of patients using a patient portal at a tertiary hospital in Riyadh, Saudi Arabia, focusing on engagement, usability, and patient-generated data. Methods: A descriptive cross-sectional study was conducted using an online survey distributed to 244 patients using the patient portal between September and December 2023. Data collected included sociodemographic characteristics, health literacy, internet and smartphone usage, and patient engagement with the portal. Results: Among the 244 respondents, 99.6% were smartphone users, and 85% reported using the patient portal. The most frequently used functionalities included scheduling appointments (60.1%) and viewing laboratory results. Significant associations were found between general satisfaction and perceptions of ease of login, information comprehension, and increased engagement (*p* < 0.05). High satisfaction was reported among those updating personal information (88.6%) and allergy status (78.1%) through the portal. Barriers to use included internet access limitations and privacy concerns. Age significantly influenced the need for training to enhance portal usage (*p* < 0.05). Conclusions: Patient engagement with the portal was high, indicating its potential as a tool for enhancing healthcare delivery. Improving usability, addressing identified barriers, and providing tailored training could further optimize patient engagement and utilization of health services.

## 1. Introduction 

Patient portals, which are electronic personal health records (ePHRs) integrated with electronic health records (eHRs), are considered effective tools for enhancing patient engagement [1]. EPHRs are electronic clinical applications that allow patients to access, manage, and communicate their health information in a private, secure, and confidential manner, thereby improving health outcomes, particularly in the management of chronic diseases. Patient involvement is crucial in the quest to improve healthcare service delivery by increasing both the quality of care and patient engagement [2].

EPHRs contain essential medical information, including current medications, allergies, health problems, vital signs, immunizations, and medical, social, and family health history. These comprehensive data enable patients to ensure the accuracy of the information and actively participate in their healthcare. However, the role of patients in managing and entering data in ePHRs remains limited compared to their role in viewing information. In many cases, passive use is prioritized over active use, likely because of the sensitive and complex nature of the clinical data stored in ePHRs [3].

Patients use ePHRs to ensure their records are complete and up-to-date, which in turn increases patient participation and satisfaction, as accurate medical information is essential for safe treatments and decision-making. Additionally, when patients enter, review, and update their information in their ePHR before and after hospital visits, it can help reduce the time physicians spend on data entry. This practice also promotes medication compliance, strengthens the patient–physician relationship, involves patients in decision-making, and ultimately improves clinical outcomes. Moreover, leveraging technology such as ePHRs is a key strategy for engaging patients in their self-care [3].

Research on patient portals reveals that both patients and healthcare providers show significant interest in ePHRs due to their potential to empower patients in their healthcare journey. However, the implementation of ePHRs faces several obstacles and challenges, as utilizing eHRs effectively requires a deep understanding of these barriers to fully realize the benefits of ePHRs [2]. One of the primary reasons for the failure of ePHR implementation is low utilization rates. A systematic review conducted in 2019 examined factors affecting ePHR utilization by patients, focusing on perceived usefulness, privacy and security concerns, and internet access to ePHRs. These factors are crucial for vendors and healthcare facilities to consider in order to ensure successful implementation and maximize the benefits of ePHRs. Additionally, increasing patient awareness and building their trust in the secure sharing of their information are essential steps in this process [4].

Frequent patient use of ePHRs is positively correlated with personalized, one-on-one training. However, challenges such as varying levels of health and technology literacy can hinder patient engagement. On the provider side, there is often a lack of workflow support for patient-generated data. Moreover, eHRs need to prioritize usability across a diverse patient population, focusing on both the user interface and the scope of content provided. EHRs were initially designed to serve healthcare providers rather than patients. However, patient access to portals has been shown to increase engagement [5], which, in turn, can improve patient experience and influence positive health behaviors. Enhanced patient engagement also correlates with a reduction in preventable deaths. Although there are some barriers in traditional workflows, many of these can be addressed with the use of advanced technology [6].

The Institute for Healthcare Improvement defines patient engagement as “actions that people take for their health and to benefit from care”. Other researchers view patient engagement as a critical change that is key to improving patient experiences, enhancing outcomes, and reducing healthcare costs, especially as healthcare shifts from acute inpatient settings to more patient-centered care at home [6]. In Saudi Arabia, e-health has seen rapid growth due to awareness and public outreach efforts, fostering widespread utilization. However, these efforts are still insufficient to achieve their full potential. Consumers generally believe e-health reduces costs and improves quality [7].

Patient-generated health data (PGHD), such as outcome reporting and mobile health data, have been introduced to improve overall health outcomes. PGHD is integrated into eHRs, enabling patients to monitor their health status without needing to visit the hospital physically. This not only increases patient satisfaction and survival rates but also encourages vendors to implement PGHD and remote monitoring functionalities, allowing both patients and caregivers to enter data [8]. However, the influx of PGHD has added to physicians’ responsibilities [9]. The increase in data has led to the involvement of artificial intelligence in EHR data management, particularly in critical care, where data volume is especially high [10].

In a study conducted in Saudi Arabia considering patient background by including patients exposed to the Internet, the result shows that ease of use is affected by facilitating conditions such as users’ knowledge, abilities, and support offered by the organization, which influenced the perception of the usefulness of electronic medical records (EMRs) and directly shaped patients’ attitudes toward using them. Among the top three ethical issues, patients, their families, and clinical staff in Saudi Arabia list confidentiality as one of the major factors [11].

Generally, even the EMRs that are utilized by healthcare providers face challenges affecting their implementation, administration, and technology factors, which greatly impacts implementation success. Additionally, users require proper training that considers both content and duration, in order to ensure they receive the benefits of the training. Particularly, human factors like resistance play an essential role in the implementation stage [12]. This study aimed to evaluate the experiences of patients using a patient portal at a tertiary hospital in Riyadh, Saudi Arabia, focusing on engagement, usability, and patient-generated data.

## 2. Methods

### 2.1. Study Design

This study employed a descriptive cross-sectional design utilizing an online questionnaire. The design was chosen to collect comprehensive information and develop a robust understanding of the research problem’s scope.

### 2.2. Study Area/Setting

The study was conducted among patients using the e-services portal of a tertiary hospital in Riyadh between 24 September and 13 December 2023. This secure online platform offers 24/7 access to personal health information through web-based or smartphone applications. Users can manage appointments, view medical reports, contact relatives, request medication refills, and update personal details via the portal. Access is restricted to patients unless they grant permission to others and includes features for managing dependent accounts for those under 18 years old. The application was developed in-house by the organization and integrated with EMRs. Multiple teams are involved to provide support to patients and technical support if needed. Additionally, the design and services are frequently updated and modified to enhance the service.

### 2.3. Study Subjects

Patients using the patient portal were selected for the study. Patients who were not currently using the application or did not have access were excluded.

### 2.4. Sample Size and Sampling Techniques

A convenience sampling method was used. The researcher visited outpatient areas in Riyadh to distribute the survey by sharing a barcode to direct the participants to the survey. The target population consisted of users of the patient portal at a tertiary hospital in Riyadh. Using the Roasoft sample size calculator, with a standard estimated population size of 20,000 users, a 95% confidence level, and a 5% margin of error, the minimum recommended sample size was determined to be 377 subjects.

### 2.5. Data Collection Tool

The questionnaire was administered using the REDCap. Platform. The online survey link was shared with participants in outpatient clinic waiting areas, and an invitation barcode for the study was distributed in these areas. Before starting the questionnaire, participants were required to confirm that they were using the application. A response was mandatory before proceeding to the next section. The researcher obtained authorization from the questionnaire developer and copyright holder [13] to use the Arabic and English versions of the questionnaire, with 17 questions removed to tailor it to the study’s needs.

The first section of the questionnaire (sociodemographic information) included nine questions about each participant’s background, including gender, age, educational level, employment status, region of residence, income, nationality health condition, and overall satisfaction with the organization’s services. The second section contained five questions assessing each participant’s health literacy. The third section included four questions about their internet and smartphone usage. The fourth section comprised nine questions related to the current application. The final section focused on patient engagement and patient-generated data, including twelve additional questions to meet the study’s objectives and evaluate the current functionality of the patient portal.

The validation process involved assessments by nine experts from a broad range of disciplines, including health informatics. A panel of experts was formed, and they were contacted via email with information about the study. Validation forms and questions were sent to the specialists via email to evaluate the clarity and relevance of the survey.

The content validity index (CVI) was used to test the questionnaire’s validity. A four-point scale was employed: 1 = strongly disagree, 2 = disagree, 3 = agree, and 4 = strongly agree. Ratings of 1 and 2 were scored as 0, while ratings of 3 and 4 were scored as 1. Two types of CVI were calculated: item-level CVI (I-CVI) and scale-level CVI (S-CVI). The S-CVI was calculated by averaging the I-CVI scores across all items on the scale (S-CVI/Ave) using the following formula: S-CVI/Ave = (sum of I-CVI scores)/(number of items) where I-CVI = (number of experts who agreed on an item)/(total number of experts). The acceptable CVI for nine experts is at least 0.78 [14]. The questionnaire demonstrated excellent content validity, with an S-CVI/Ave of 0.89.

The web-based survey was pilot-tested, with 24 participants responding. No comments were received, and all participants indicated that the survey was clear. The internal consistency of the questionnaire was measured using Cronbach’s alpha, which was 0.90, demonstrating good internal reliability.

### 2.6. Data Collection

Participants were invited via a barcode that directed them to the survey link. All individuals who agreed to participate and were using the application were included in the sample. The participants were recruited using a convenience sampling technique, with no predetermined sampling frame. Participant identity was not recorded, ensuring confidentiality. Upon completing the survey, participants were shown a thank-you message. No incentives were offered for survey completion.

### 2.7. Statistical Analysis

The demographic characteristics of the participants were reported using descriptive statistics, such as frequencies and percentages. Categorical variables were analyzed using chi-squared tests to determine associations between demographic characteristics and other variables. The analyses were performed using Stata Statistical Software: Release 18. (StataCorp LLC, College Station, TX, USA, 2023). A *p*-value of 0.05 or less was considered statistically significant.

## 3. Results

### 3.1. Characteristics of Study Participants

The survey received 299 responses, with two individuals not agreeing to participate and 42 not using the application, resulting in an 85.4% completion rate. A total of 244 responses were analyzed. The socioeconomic characteristics of the participants are presented in Table 1. Approximately half of the respondents (53.69% n = 131) were from Riyadh. Similarly, 53.28%, (n = 130) were female, and more than half were aged 20–39 or 40–59 (49.58% n = 117 and 35.59% n = 84, respectively). Nearly half of the sample (45.90% n = 112) held bachelor’s degrees. Regarding employment status, 53.50% (n = 130) of respondents were employed.

The survey results on health status indicated that 18.03% of respondents reported having cardiac disease, while 13.93% had diabetes. Hypertension was reported by 10.66% of participants, and 5.33% had either asthma or COPD. Cancer was also prevalent among 5.73% of the sample. Less common conditions included renal failure (4.51%), sickle cell disease (1.64%), and psychiatric conditions (1.23%) (Figure 1).

### 3.2. Internet Use and Online Health Information Seeking Behavior

Nearly all participants were smartphone users (99.59% n = 242), with 90.16% (n = 220) using their phones multiple times a day to go online. More than half of the participants (60.58% n = 146) used the Internet to search for health-related information. Additionally, a high percentage had health applications on their mobile phones (88.89% n = 216).

### 3.3. Health Literacy

Most participants indicated that they could read hospital materials without assistance (78.19% n = 190) and fill out hospital forms independently (78.19% n = 173). Furthermore, 72.13% (n = 174) reported no difficulty in learning about their medical condition due to understanding written information. However, a lower proportion of participants felt confident in evaluating health-related information online (45.42% n = 109) and communicating reliable health information to others (55.41% n = 133).

### 3.4. Satisfaction with Healthcare

A significant proportion of patients were either satisfied or very satisfied with the service provided by the hospital (84.36%, n = 205), as illustrated in Figure 2.

### 3.5. Patient Portal

The majority of users had been using the application for more than a year (70.04% n = 166). However, only a few participants reported daily utilization (5.76% n = 14). Among the reasons for using the application, scheduling appointments was the most utilized functionality (60.07% n = 179). Notably, about a quarter of the sample disagreed or strongly disagreed with the statement that logging into the patient portal was easy (26.36% n = 63).

### 3.6. Using the Application for Data Entry

Currently, the application allows patients to enter personal information. Of the participants, 30.38% (n = 72) updated their personal information via the application, with a high satisfaction rate of 88.57% (n = 62). Additionally, 14.04% (n = 33) used the application to update their allergy status, with 78.12% (n = 25) expressing satisfaction. A minority of participants had medical conditions requiring monitoring and documentation on paper (e.g., glucose level, blood pressure) (14.64% n = 35) Meanwhile, the majority believed that documenting through the application would be easy for them (83.54% n = 198). The minority who found it difficult cited a few barriers, with internet access receiving the highest score (18.98% n = 15) as shown in Figure 3. This may explain why most participants agreed that the application should provide access to information offline (59.48% n = 138).

### 3.7. Desired Application Features

Most of the participants agreed that the application should include access information ahead of medical appointments (76.17% n = 179). Additionally, tracking prescribed medications and exercises, both in the hospital and during outpatient rehabilitation, was supported by 69.1% (n = 161). Furthermore, 68.36% (n = 160) of participants wanted to share access to the mobile health app with caregivers and family members, as shown in Figure 4.

### 3.8. Patient-Required Training and Ease of Data Entry Process

Participants showed a strong interest in utilizing the application if provided with the required training (79.7 n = 192). The study found a significant association between age and the need for training (*p* < 0.05) as indicated in Table 2, but not with literacy, as shown in Table 3.

## 4. Discussion

This study focused on users of the patient portal at a tertiary hospital, examining participants’ demographic, literacy, technology utilization, and online behavior. Additionally, it explored participants’ knowledge levels, utilization of the patient portal, and the potential for more engaging functionality.

The results revealed that nearly all participants were smartphone users (99.59%), a figure closely aligned with findings from a 2020 study titled “Adoption of a Personal Health Record in the Digital Age: Cross-Sectional Study” conducted at MNGHA Riyadh, which reported a smartphone user rate of 92.2% among 546 participants. This earlier study also indicated a relationship between increasing age and lower application utilization [13], a finding consistent with the result of the current study. In the current study, a high proportion of participants using the application fell within the 18–39 age group. Additionally, the highest proportion of users were likely to be married, university graduates, and employed. Health literacy was higher among females and was notably better in Riyadh than outside the city. Moreover, a significant majority of participants were satisfied or very satisfied with the healthcare service provided by the organization.

Based on our research, we found that patients are more comfortable receiving medical information rather than providing it themselves. This may limit their use of the portal for self-data entry and make them prefer in-person visits for verbal information sharing with healthcare providers.

A systematic analysis titled “Discovering Patient Portal Features Critical to User Satisfaction” found that patient satisfaction with applications is closely associated with the ease of scheduling appointments [15]. In the current study, 26.36% of participants disagreed or strongly disagreed with the statement that logging into the patient portal was easy, indicating that difficulty or an unclear process during access could negatively impact user satisfaction. This indicates the need to initiate a plan to support the patient with the login and registration process since this is the initial step to start a patient’s portal journey.

Further supporting this, a study on the benefit of SMS within patient portals for diabetic patients found that enhancing communication through the portal positively influenced patient satisfaction with care [16]. The current study also indicates a relationship between general satisfaction with healthcare services and agreement with statements such as “information in the patient portal is understandable”, “logging into the patient portal is easy”, and “using the patient portal makes me feel more involved in my care”. As highlighted in previous research, patient satisfaction and ease of use are related [15], underscoring the need to review the steps and ensure the application is user-friendly. Patient satisfaction has a huge impact on patient acceptance of organization services, and thus plays a key role in patient portal empowerment.

The patients reported strong interest in using the application, with scheduling appointments being the most frequently used functionality, followed by viewing laboratory results. Additionally, the current application functionalities include data documentation by the patients, such as updating allergies and personal information. Among those who reported using these features, satisfaction was high: 88.57% for updating personal information and 78.12% for updating allergy status. These high satisfaction rates may reflect the ease of use, as previously noted, where satisfaction is closely linked to usability.

The majority of participants had positive attitudes toward the patient portal, particularly regarding data documentation, with 83.54% believing that documentation through the application would be easy. Our findings indicate that managing appointments is the most frequently used functionality, consistent with a similar study conducted in Saudi Arabia, which reported even higher utilization rates [13]. Conversely, managing personal health information was among the least utilized features.

A few barriers were identified, with the most common being internet access, which is explained by their concern about a need to access information in offline use cases. Other barriers were identified, such as difficulties in using the application and privacy concerns. A systematic review investigating factors affecting the use of ePHRs among patients categorized these barriers under human–technology factors, emphasizing the importance of incorporating patient perspectives in the design process [4]. Privacy concerns, in particular, were highlighted by participants, a barrier also noted in multiple studies due to the potential of shared device usage [17]. A study in Saudi Arabia found that 79.3% of participants were concerned about the privacy of their health information, prompting recommendations for stronger regulations to address these issues [18].

The study also found that individuals with varying levels of health literacy were more likely to use the application for data documentation if provided with appropriate training. A significant association was identified between age and ability to use the portal if the needed training was provided (*p* < 0.05). A study conducted in Saudi Arabia, using a simulated PHR, demonstrated that those who received video support performed better in completing tasks within the portal compared to those who did not receive such support [11]. Training need not be limited to classroom settings, as patients can receive instructions via multiple channels, such as interactive videos [19].

One of the main limitations of this study was the small sample size, due to the response rate and restrictions on data collection being limited to the outpatient area by organization regulations. Additionally, there is a possibility of bias due to the use of self-reported data. Future studies should aim to overcome these limitations by performing a more comprehensive statistical analysis using application utilization records and including data from inpatient and specific specialty clinics.

### Conclusions

The results of this study demonstrate significant interest in using the studied patient portal; however, patient awareness and adoption need to be improved. Moreover, the majority of participants expressed interest in utilizing the application for documentation, with only a minority citing barriers. Improving the patient portal’s usability and addressing barriers such as internet access and privacy concerns can significantly enhance patient engagement. This, in turn, will support the Saudi government’s e-Health initiative to provide more accessible and effective healthcare services. Future research should include the perspectives of healthcare providers and application data logs to provide a more comprehensive understanding.

## 5. Declarations

I declare that this thesis, presented in this research project report, entitled “Feasibility of Using Patient Portal to Enhance Patient Engagement and Patient-Generated Data in Tertiary Hospital”, submitted in partial fulfillment of the requirements for the degree of Master of Health Informatics at King Saud bin Abdulaziz University for Health Sciences, has been composed solely by myself. It has not been submitted, in whole or in part, for any previous application for a degree. The work presented is entirely my own and was conducted under the guidance and supervision of Dr. Abdullah Alanazi.

## Figures and Tables

**Figure 1 healthcare-13-00518-f001:**
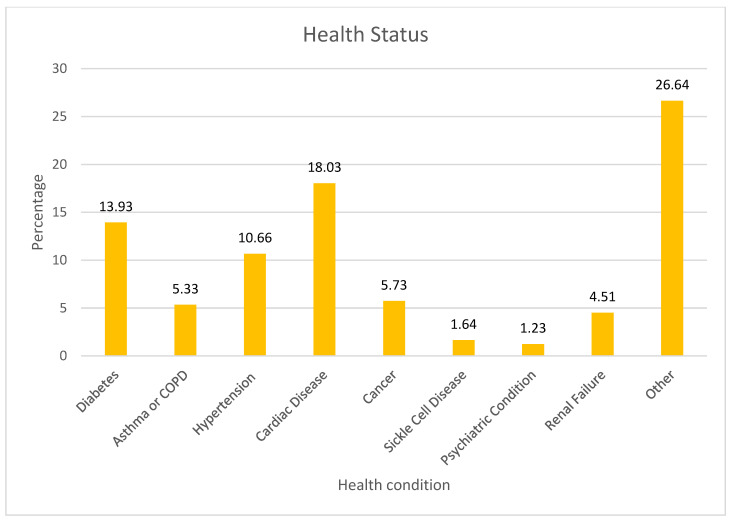
Health status.

**Figure 2 healthcare-13-00518-f002:**
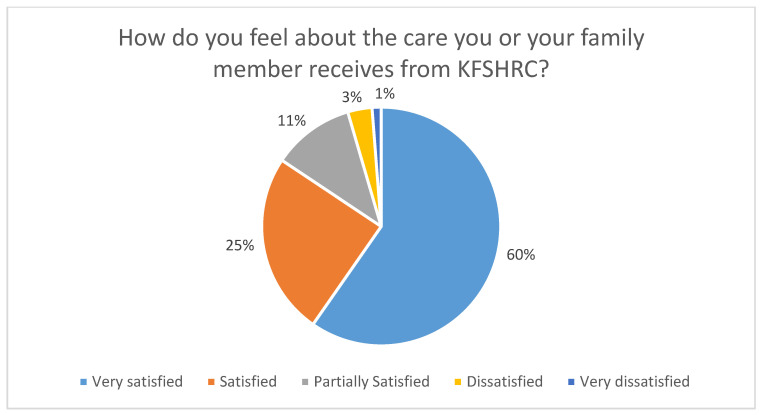
How do participants feel about the care provided by the organization?

**Figure 3 healthcare-13-00518-f003:**
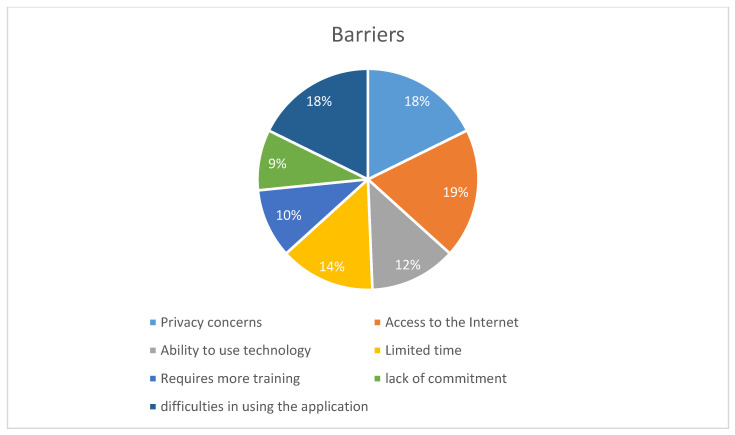
Things that may hinder a person from using the application to enter medical information.

**Figure 4 healthcare-13-00518-f004:**
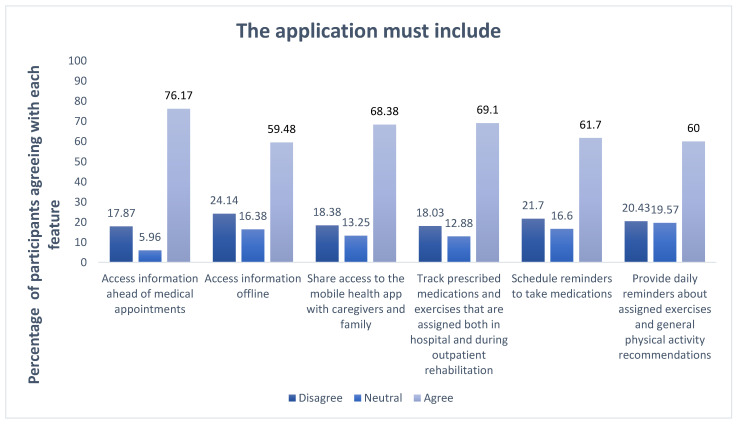
Desired application features.

**Table 1 healthcare-13-00518-t001:** Participants’ demographic data.

Do You Use the Patient Portal?	Frequency	Percent
Yes	244	85.31
No	42	14.69
Where do you live in Saudi Arabia?		
Riyadh	131	53.69
Outside Riyadh	113	46.31
Age groups		
0–19	11	4.66
20–39	117	49.58
40–59	84	35.59
60–79	22	9.32
80–99	2	0.85
Gender		
Male	114	46.72
Female	130	53.28
Marital status		
Married	153	62.70
Single	75	30.74
Divorced	11	4.51
Widowed	5	2.05
Highest grade		
Elementary school or less	20	8.20
Middle school	5	2.05
High school	57	23.36
University	112	45.90
Postgraduate	50	20.49
Employment status		
Retired	33	13.58
Unemployed	52	21.40
Employed	130	53.50
Student	19	7.82
Self-Employment	9	3.70
Monthly income		
<5000 SAR month	82	34.45
5000–9999 SAR month	36	15.13
10,000–19,999 SAR month	77	32.35
20,000–49,999 SAR month	38	15.97
>50,000 SAR month	5	2.10

**Table 2 healthcare-13-00518-t002:** Training and age.

Age	If You Are Provided with the Required Training, Will You Use the Application for Documentation?				*p*-Value	Pearson Chi^2^
	Yes	No	Sometimes	Total		
0–19	6 (3.23%)	0 (0.00%)	4 (13.33%)	10 (4.33%)	0.0059	21.50
20–39	99 (53.23%)	4 (26.67%)	13 (43.33%)	116 (50.22%)		
40–59	65 (34.95%)	8 (53.33%)	8 (26.67%)	81 (35.06%)		
60–79	16 (8.60%)	2 (13.33%)	4 (13.33%)	22 (9.52%)		
80–99	0 (0.00%)	1 (6.67%)	1 (3.33%)	2 (0.87%)		
Total	186 (100.00%)	15 (100.00%)	30 (100.00%)	231 (100.00%)		

**Table 3 healthcare-13-00518-t003:** Training and health literacy.

If You Are Provided with the Required Training, Will You Use the Application for Documentation? [Literacy]					*p*-Value	Pearson Chi^2^
	Low	Mid	High	Total	0.3655	4.31
Yes	74 (74.75%)	52 (81.25%)	59 (85.51%)	185 (79.74%)		
No	8 (8.08%)	6 (9.38%)	3 (4.35%)	17 (7.33%)		
Sometimes	17 (17.17%)	6 (9.38%)	7 (10.14%)	30 (12.93%)		
Total	99 (100.00%)	64 (100.00%)	69 (100.00%)	232 (100.00%)		

## Data Availability

The dataset is available from the authors upon reasonable request and after approval from the Hospital.

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
