# Peer review of "Feasibility of Using Patient Portal to Enhance Patient Engagement and Patient-Generated Data in Tertiary Hospital"

_healthcare, 2025, doi:10.3390/healthcare13050518_

Round 1

Reviewer 1 Report

Comments and Suggestions for Authors

Dear authors,

thank you for having the opportunity to read your paper in advance.

All in all, I have some central concerns. What is the value added in your survey? Yes, we know that xy% do this or that, but the paper is missing a clear analytical mean. You also have not integrated any acceptance model, so your conclusion is quite superficial.

From a methodological point of view, I also have one great concern, you do an online-survey and ask the people if the use the internet? As we know, the external validity of online-surveys is always lower than others, but in your case it is one of the central limitations and excludes important insights.

So, what is the theoretical lense, what really can be the aim und discuss clearly what you can say with your methods.

Author Response

Reviewer 1

Comments and Suggestions for Authors

Dear authors,

thank you for having the opportunity to read your paper in advance.

All in all, I have some central concerns. What is the value added in your survey? Yes, we know that xy% do this or that, but the paper is missing a clear analytical mean. You also have not integrated any acceptance model, so your conclusion is quite superficial.

we are trying to evaluate patient experience toward the patient portal to enhance patient engagement and patient-generated data in a tertiary hospital check the possibility of adding functionality to improve patient-generated data and engagement more than being the receiver, in other words, increase active use to not limit the application to passive use. More analysis was added to the conclusion.

From a methodological point of view, I also have one great concern, you do an online-survey and ask the people if the use the internet? As we know, the external validity of online-surveys is always lower than others, but in your case it is one of the central limitations and excludes important insights.

The survey was distributed as a barcode directing the participant to the survey link which doesn’t indicate being an active internet user, there was no question about whether participants use or not but the question concerns their usage and behavior toward the internet. 

So, what is the theoretical lense, what really can be the aim und discuss clearly what you can say with your methods.

The survey was distributed as a barcode directing the patient to the online survey which doesn’t indicate the utilization details, there was no question if the participant used the internet or not, but a few questions related to internet behavior and the  smartphone since  the application is smartphone and web-based access.

Reviewer 2 Report

Comments and Suggestions for Authors

Dear Authors,

I have a few points regarding your paper that need to be addressed:

Minor Issues:

  1. There is no need to include official documents of approval/consent in the appendices. Please upload these as ‘Unpublished Documents’ instead.
  2. Several sections are missing, including Section 1.3.3, Author Contributions, and Funding (refer to page 15). Additionally, the Acknowledgment section should omit thank-you statements to family members, as they are not appropriate here.
  3. Since your study focuses on the feasibility of using an electronic system or application, the tables should exclusively present results relevant to this aspect.

Major Issues:

  1. Separate the Introduction section from the Literature Review. The Introduction should outline the importance and motivation of the topic, while the Literature Review should detail at least five studies related to the topic, particularly those conducted in Saudi Arabia over the past five years. For reference, consider the following source that discusses the failure of electronic health record systems in Saudi Arabia:                                                     "Healthcare Software Design and Implementation—A Project Failure Case,” Journal of Software: Practice and Experience, 50(7), 2020.
  2. Change the title of the paper to “The Feasibility of Using the Altakhassusi Application…” as per the formal approval. This title more accurately reflects the content of your document. The current version does not clearly define the acceptability and usability of the application, nor does it explain how these aspects are measured.
  3. In Table 2 (Health Status), if the total number of people with a disease is 128, please clarify the ‘If yes’ rows. Specifically, if the total number of people is 128, how can the number of people with Diabetes be listed as 264?
  4. Add a paragraph describing the Altakhassusi application (referred to as the patient portal), including its design, technical environment, and other relevant details.

Thank you for your attention to these matters.

Best regards,

Author Response

Comments and Suggestions for Authors

Dear authors,

Reviewer 2

Comments and Suggestions for Authors

Dear Authors,

I have a few points regarding your paper that need to be addressed:

Minor Issues:

  1. There is no need to include official documents of approval/consent in the appendices. Please upload these as ‘Unpublished Documents’ instead.

Done and removed

  1. Several sections are missing, including Section 1.3.3, Author Contributions, and Funding (refer to page 15). Additionally, the Acknowledgment section should omit thank-you statements to family members, as they are not appropriate here.

Done removed

  1. Since your study focuses on the feasibility of using an electronic system or application, the tables should exclusively present results relevant to this aspect.

Few tables removed

Major Issues:

  1. Separate the Introduction section from the Literature Review. The Introduction should outline the importance and motivation of the topic,

Per the journal guidance, it should be combined. However, it changed and separated as requested

  1. while the Literature Review should detail at least five studies related to the topic, particularly those conducted in Saudi Arabia over the past five years. For reference, consider the following source that discusses the failure of electronic health record systems in Saudi Arabia:                                                     "Healthcare Software Design and Implementation—A Project Failure Case,” Journal of Software: Practice and Experience, 50(7), 2020.

Included with one more study in Saudi Arabia

Two more study cited including the suggested one

  1. Change the title of the paper to “The Feasibility of Using the Altakhassusi Application…” as per the formal approval. This title more accurately reflects the content of your document. The current version does not clearly define the acceptability and usability of the application, nor does it explain how these aspects are measured.

Done, however, the primary title doesn’t reflect the content and the aim. For a new title used I consider removing the application name and hospital name to be

(Evaluate Patient Experience Toward Patient Portal to Enhance Patient Engagement and Patient-Generated Data in a Tertiary Hospital)

  1. In Table 2 (Health Status), if the total number of people with a disease is 128, please clarify the ‘If yes’ rows. Specifically, if the total number of people is 128, how can the number of people with Diabetes be listed as 264?

Done as by chart

  1. Add a paragraph describing the Altakhassusi application (referred to as the patient portal), including its design, technical environment, and other relevant details. Thank you for your attention to these matters.

Done

Round 2

Reviewer 2 Report

Comments and Suggestions for Authors

Major Comments:

1. Clarification on Terminology: Could you please clarify what "EMR" stands for on page 3/12?

2. Title Concerns: In my previous major comment (No. 2), I suggested changing the title of the paper as it does not accurately reflect the content. Unfortunately, the revised title still suffers from this issue. I would appreciate it if you could either adopt my earlier suggestion or propose a new title that better encapsulates the paper's focus. Additionally, the term “Portal” is used in the title, whereas the document refers to it simply as an application. In my earlier comment (No. 5), I requested a paragraph detailing the Altakhassusi application. While your response stated “Done,” it would have been helpful to include the relevant section or subsection number for quicker reference. I located the mention in Section 3.2, titled “Study Area/Setting,” which describes the services the application offers to patients. However, I was expecting a more technical description and design details of the application. Because the authors might not have specialized in IT, they could obtain such data from the developers or the IT department at the hospital.

3. Response to Previous Comments: Regarding my major comment (No. 4), your response of “Done as by chart” does not adequately address my inquiry. I did not request the removal of the table or its replacement with a chart; rather, I posed a direct question that needs clarification.

   Additionally, I noted that the total number of individuals with the disease is 128, while those without it total 92, resulting in 220 participants in the study. However, on page 5/12, you state that the total number of individuals whose responses were analyzed is 244. Could you clarify how this discrepancy arose? Does this imply that 24 responses are unaccounted for? Please provide ME with an explanation.

Author Response

Clarification on Terminology: Could you please clarify what "EMR" stands for on page 3/12?

EMR stands for Electronic Medical Record and will include that in the manuscript beside the abbreviation

Title Concerns: In my previous major comment (No. 2), I suggested changing the title of the paper as it does not accurately reflect the content. Unfortunately, the revised title still suffers from this issue. I would appreciate it if you could either adopt my earlier suggestion or propose a new title that better encapsulates the paper's focus.

The title corrected as requested but the application and organization names are removed

Additionally, the term “Portal” is used in the title, whereas the document refers to it simply as an application. In my earlier comment (No. 5), I requested a paragraph detailing the Altakhassusi application. While your response stated “Done,” it would have been helpful to include the relevant section or subsection number for quicker reference. I located the mention in Section 3.2, titled “Study Area/Setting,” which describes the services the application offers to patients. However, I was expecting a more technical description and design details of the application. Because the authors might not have specialized in IT, they could obtain such data from the developers or the IT department at the hospital.

more details were added under “Study Area/Setting,”.However, the technical team are sharing a few superficial information

Response to Previous Comments: Regarding my major comment (No. 4), your response of “Done as by chart” does not adequately address my inquiry. I did not request the removal of the table or its replacement with a chart; rather, I posed a direct question that needs clarification.

After reviewing the table, we found out it represents accurate numbers. However, its corrected in the new figure for better data presentation.

   Additionally, I noted that the total number of individuals with the disease is 128, while those without it total 92, resulting in 220 participants in the study. However, on page 5/12, you state that the total number of individuals whose responses were analyzed is 244. Could you clarify how this discrepancy arose? Does this imply that 24 responses are unaccounted for? Please provide ME with an explanation.

 there was a missing response from 24 participant in this part